

# The relationship between numerosity perception and mathematics ability in adults: the moderating role of dots number

Ji Sun[1,2] and Pei Sun[2]

[1] School of Education Science, Anshun University, Anshun, Guizhou, China
[2] Department of Psychology, Tsinghua University, Beijing, Beijing, China

## ABSTRACT

**Background**. It has been proposed that numerosity perception is the cognitive underpinning of mathematics ability. However, the existence of the association between numerosity perception and mathematics ability is still under debate, especially in adults. The present study examined the relationship between numerosity perception and mathematics ability and the moderating role of dots number (*i.e.,* the numerosity of items in dot set) in adults.

**Methods**. Sixty-four adult participants from Anshun University completed behavioral measures that tested numerosity perception of small numbers and large numbers, mathematics ability, inhibition ability, visual-spatial memory, and set-switching ability.

**Results**. We found that numerosity perception of small numbers correlated significantly with mathematics ability after controlling the influence of inhibition ability, visual-spatial memory, and set-switching ability, but numerosity perception of large numbers was not related to mathematics ability in adults.

**Conclusions**. These findings suggest that the dots number moderates the relationship between numerosity perception and mathematics ability in adults and may contribute to explaining the contradictory findings in the previous literature about the link between numerosity perception and mathematics ability.

## INTRODUCTION

Mathematics ability is a unique cognitive ability for humans, which is essential to our everyday life. Yet, there is a subgroup of people who have trouble acquiring mathematics skills. What factors influence mathematics ability? Previous studies have found that many domain-general factors could predict individual differences in mathematics ability, including memory (*Raghubar, Barnes & Hecht, 2010*), executive function (*Kroesbergen et al., 2009*) and early language ability (*Hecht et al., 2001*). In addition, symbolic number system (SNS) ability was an essential domain-specific predictor of mathematical ability (*Lau et al., 2021*; *Lyons et al., 2018*). For example, *Geary & Vanmarle (2016)* found that young children's core symbolic abilities significantly correlated with later mathematics

Corresponding author
Pei Sun, peisun@tsinghua.edu.cn

achievement. Recently, several studies have found a potential role of numerosity perception in facilitating individuals' formal mathematical skills (*e.g., Elliott et al., 2019*; *Halberda, Mazzocco & Feigenson, 2008*; *Mazzocco, Feigenson & Halberda, 2011*; *Park & Brannon, 2013*). Furthermore, numerosity perception has been found to be a more important predictor of mathematics ability than other general cognitive functions (*Chen & Li, 2014*; *Halberda, Mazzocco & Feigenson, 2008*).

Numerosity perception is the elementary numerical ability in both humans and animals, representing nonverbal information of quantity without counting (*Dehaene, 1997*; *Dehaene & Changeux, 1993*; *Feigenson, Dehaene & Spelke, 2004*). A numerosity discrimination task is commonly applied to measure the precision of numerosity perception in which two dot sets are displayed briefly, and the participants are asked to judge which set is more numerous (*Halberda & Feigenson, 2008*; *Inglis et al., 2011*). The numerosity discrimination performance, such as weber fraction ($w$), was used as an index of numerosity perception. The $w$ is an abstract measure of discrimination threshold that specifies the precision of the numerosity representation and represents the width of numerosity tuning curve (*Kersey & Cantlon, 2017*; *Piazza et al., 2004*), where higher $w$ reflects less accurate numerosity representation.

The supporting evidence for the relationship between numerosity perception and mathematics ability came from the fact that numerosity discrimination performance (which reflects numerosity perception) correlated with mathematics ability (*Halberda, Mazzocco & Feigenson, 2008*; *Libertus, Feigenson & Halberda, 2011*; *Mazzocco, Feigenson & Halberda, 2011*), which was measured by different tasks (*Holloway & Ansari, 2009*; *Fagerlin et al., 2007*). Furthermore, it has been shown that training on numerosity perception tasks improved formal mathematics abilities in both children and adults (*Hyde, Khanum & Spelke, 2014*; *Park & Brannon, 2013*). However, it should be noted that many studies did not find such a relationship (*Gilmore et al., 2013*; *Lyons & Beilock, 2011*; *Lyons et al., 2014*). The inconsistent results might ascribe to two reasons. First, the inconsistency might be caused by age differences of participants recruited (*Gilmore et al., 2013*; *Inglis et al., 2011*; *Schneider et al., 2017*). For instance, *Inglis et al. (2011)* found that the association between numerosity perception and mathematics skills only existed in children but not in adults. Second, a methodological difference which refers to the different numerosity of items in the dot set (*i.e.,* dots number) used in previous studies, might also contribute to the inconsistency. There is evidence that the number of stimuli that has to be estimated (*i.e.,* dots number) could influence the perception mechanism (*Anobile et al., 2016*; *Anobile, Cicchini & Burr, 2014*; *Anobile, Cicchini & Burr, 2016*; *Pome et al., 2019*; *Fornaciai & Park, 2017*; *Zimmermann, 2018*). It seems that small numbers of stimuli (relatively sparse dot-pattern) were sensed by the numerosity mechanism, but large numbers (relatively dense dot-pattern) were sensed by the density-texture mechanism (*Anobile et al., 2016*; *Anobile, Cicchini & Burr, 2014*; *Zimmermann, 2018*). Actually, previous studies found that only the numerosity perception of small numbers correlated with mathematics ability in children (*Anobile et al., 2016*).

Given the above, this study aimed to investigate the relationship between numerosity perception and mathematics ability and the effect of dots number on this relationship in

adults. In particular, we conducted three measurements to assess participants' mathematics ability, including an arithmetical test, a subjective mathematics ability evaluation, and their mathematics course scores. In the meantime, we conducted the numerosity discrimination tasks of small and large number conditions separately, which allowed us to evaluate the numerosity perception of different dot numbers. In addition, to control the potential effects of other general cognitive abilities on the relationship between numerosity perception and mathematics ability (*Fuhs & McNeil, 2013*; *Gilmore et al., 2013*), all participants performed three measurements on their inhibition ability, visual-spatial memory, and set-switching ability. We predicted that the number of dots would moderate the relationship between numerosity perception and mathematics ability in adults, and only the numerosity perception of small numbers would be related to mathematics ability.

## MATERIALS & METHODS

### Participants

Sixty-four undergraduate students from Anshun University (28 males and 36 females, $M$ age = 20.27, $SD$ age = 1.11) participated in this study and received financial compensation for their time. This study was approved by the ethics committee of the School of Education Science, Anshun University (ID number: ASU-JYXY-201903) and all participants provided written informed consent at the beginning of the experiment. Three participants were excluded from the analyses due to either misunderstanding experimental task demand or missing data caused by computer errors.

### Apparatus and software

All measurements were completed in a dimly lit room. Computer-based experiments were completed using MATLAB (R2016b; MathWorks, Cambridge, MA, USA) and PsychToolbox (*Brainard, 1997*) on a 23-inch LED monitor (Dell: U2312 HM) with 1,600 × 1,080 resolution at 60 Hz and a standard desk (width: 70 cm; length: 150 cm), viewed binocularly from a distance of 60 cm.

### Measures

Numerosity perception, mathematics ability, inhibition ability, visual-spatial memory, and set-switching ability were measured in the following sequential order. The whole experimental procedure took approximately 85 min.

### Numerosity perception

A modified version of a numerosity discrimination task adapted from *Anobile, Cicchini & Burr (2014)* was used to measure numerosity perception. Two sets of dots were of equal size and presented simultaneously for 250 ms on either left or right side of a central fixation point. Each set was constrained to a 14°-diameter virtual circle and comprised a number of 0.3° (in visual angle) diameter dots, half white and half black. For each trial, the set on the right side of the fixation point was the reference, and the left set was the probe. In separate blocks, the reference always contained 11 dots (small number condition) or 152 dots (large number condition), while the numerosity of the probe was changed from trial to trial,

following the Method of Constant stimuli that varied the number of dots from 50% to 200% of the number of the reference that was split into seven equal log unit steps. For small number condition, the probe patches contained 6, 7, 9, 11, 14, 17, or 22 dots. For large number condition, the probe patches contained 76, 96, 121, 152, 192, 241, or 304 dots. Each of the seven probe stimulus levels was tested for 16 trials, and each block consisted of 112 trials. For each trial, participants were asked to judge as accurately and quickly as possible which set had more dots. Two blocks were presented in counterbalanced order. Participants gave their responses using a keyboard. Based on the pilot study, 14 practice trials were given to participants with feedback before the formal data collection.

"The proportion of probe greater trials" was plotted against the reference number and fitted with the Methods of Probits (*Sun et al., 2013*). The 50% point estimated the point of subjective equality (PSE), and the difference in numerosity between the 25% and 75% points provided the just notable difference (JND), which was then divided by PSE to estimate the Weber fraction ($w$).

## Mathematics ability

Three measures were used to assess participants' mathematics ability, including a modified-version arithmetical test (AT), a subjective mathematics ability evaluation, and the mathematics course score. The arithmetical test, which was designed based on *Shalev et al. (2001)*, assessed basic arithmetic skills. The types of items that were tested included number facts, complex arithmetic, decimals calculation, and fraction calculation. In the self-developed measure of subjective mathematics ability evaluation, participants were instructed to evaluate their mathematics ability compared with other people using a 10-point scale (1 = the worst level, 10 = the best level). Furthermore, participants' college entrance examination score was used as the mathematics course score. Following the method of convention of *Holloway & Ansari (2009)*, the scores for the three tasks were converted into z-scores and averaged to create an overall mathematics ability score for each participant, which was then used in the following analyses as an index of participants' mathematics ability.

## Inhibition ability

Inhibition ability is the ability to suppress dominant, automatic, or prepotent response for irrelevant or no-longer-relevant information (*Toll et al., 2011*). The classical Go/No-go task was used to assess inhibition ability (*Falkenstein, Hoormann & Hohnsbein, 1999*). In this task, participants were told to respond to a yellow square (Go stimuli) and stop responding to a yellow circle (No-go stimuli). Participants were given five practice trials with feedback. The official task had 200 test trials without feedback, including 150 Go trials and 50 No-go trials. For each participant, the accuracy for No-go trials was recorded as the inhibition score.

## Visual-spatial memory

Visual-spatial memory is a system for retaining location and object information (*Wood, 2011*). To assess participants' visual-spatial memory ability, a widely used memory span task for visual patterns was administered (*Mejias, Gregoire & Noel, 2012*). The task was
developed by *Wilson, JH & Power (1987)* and consisted of the presentation of boxes with some boxes being filled in. The participants were instructed to remember the pattern of the boxes, which was displayed for 2 s. After a blank interval of 2 s, the boxes would be presented again with one box missing. The participants had to recall which box was missing. The initial pattern involved filling in two boxes, and upon receiving two correct responses out of three attempts, the complexity of the pattern would increase. The final number of successfully recalled boxes was recorded as a memory span score.

### Set-switching ability

Set-switching ability is the ability to direct actions and thoughts to selected goals; it has been thought of as a kind of executive control. In this study, a modified Trail Making Test was presented in the paper-and-pencil version to measure set-switching ability (*Arbuthnott & Frank, 2000*). The test consisted of three trials: Part A-1, Part A-2, and Part B. Part A-1 included drawing a line connecting consecutive numbers from 1 to 25. Part A-2 included drawing a line connecting consecutive letters from A to Y. Part B included drawing a line connecting alternating numbers and letters in sequence (*i.e.,* 1-A-2-B and so on). Participants were shown three trials in order. In each trial, a timer was set when the participants began drawing a line on paper and stopped when they put the pencil on the desk. The time to complete each trial was recorded. The ratio of completion time between Part B and Part A (average completion time of Part A-1 and Part A-2) was calculated as a set-switching score. A higher set-switching score indicated a poorer set-switching ability.

### Data analysis

The statistical analyses of data were performed using SPSS software (Version 23.0). The correlations between all variables were measured using Zero-order correlation analyses. Then, we used hierarchical regression analysis to explore the predictors of mathematics ability.

## RESULTS

### Mathematics ability score

We looked for the correlations of the participants' performance on the three mathematics ability tests. Subjective mathematics ability evaluation was significantly correlated with AT accuracy ($r = 0.32$, $p = 0.013$) and the mathematics course score ($r = 0.32$, $p = 0.012$). However, AT accuracy was not significantly correlated with mathematics course scores, $r = 0.15$, $p = 0.257$. Given that mathematics ability is a complex construct and includes different skills (*Mannamaa et al., 2012*), these results showed that performances in three tests measured not only common aspects but also different aspects of mathematics ability. This might, in part, explain the previous contradictory relationships between numerosity perception and mathematics ability. To provide an evaluation of overall mathematics ability, we followed *Holloway & Ansari (2009)* and used the mathematics ability score for each participant in further analyses.
**Table 1  Descriptive statistics and Zero-order correlations.**

|  | Range | M | SD | 1 | 2 | 3 | 4 | 5 | 6 |
|---|---|---|---|---|---|---|---|---|---|
| 1. $w_{small}$ | 0.12–0.98 | 0.34 | 0.15 | 1.00 | | | | | |
| 2. $w_{large}$ | 0.11–0.89 | 0.34 | 0.17 | 0.47*** | 1.00 | | | | |
| 3. Mathematics ability | −2.47–1.29 | 0.00 | 0.71 | −0.42*** | 0.10 | 1.00 | | | |
| 4. Visual-spatial memory | 2–16 | 8.43 | 3.34 | 0.14 | −0.05 | 0.26 | 1.00 | | |
| 5. Inhibition ability | 0.64–0.96 | 0.84 | 0.08 | −0.27* | −0.15 | −0.19 | −0.03 | 1.00 | |
| 6. Set-switching ability | 0.07–5.35 | 1.75 | 0.94 | 0.08 | −0.15 | 0.11 | 0.06 | 0.00 | 1.00 |

Notes.
***$p < 0.001$.
**$p < 0.01$.
*$p < 0.05$.

## The correlations between numerosity perception and mathematics ability

A zero-order correlation matrix was calculated to investigate the relationship between all measurements (See Table 1). Results showed that mathematics ability significantly correlated with $w_{small}$ ($r = -0.42$, $p = 0.001$; the alternative Pearson $r = -0.36$, $p = 0.004$, see Fig. 1), but not with $w_{large}$ ($r = 0.10$, $p = 0.444$; the alternative Pearson $r = -0.11$, $p = 0.414$), indicating that lower mathematical ability was associated with higher $w$ (reflecting poorer numerosity discrimination precision) in the small number condition. Further correlation analysis showed that $w_{small}$ was significantly correlated with AT accuracy and the mathematics course score ($r = -0.32$, $p = 0.017$; $r = -0.37$, $p = 0.005$), but not with subjective mathematics ability evaluation ($r = -0.21$, $p = 0.125$). Conversely, we found no significant correlations between $w_{large}$ and three mathematics ability tests, $rs < 0.24$, $ps > 0.075$. It should be noted that there was also a significant correlation between $w_{small}$ and $w_{large}$ ($r = 0.47$, $p < 0.001$; the alternative Pearson $r = 0.53$, $p < 0.001$). Here, we further examined the unique effect of dots number (*i.e.*, small number and large number conditions) on the relationship between numerosity perception and mathematics ability. We used a specific test of the difference between two dependent correlations with one variable in common developed by *Lee & Preacher (2013)* to compare the correlations between $w_{small}$, $w_{large}$ and mathematics ability. Results showed that there was a significant difference between the correlation of $w_{small}$ and mathematics ability and the correlation of $w_{large}$ and mathematics ability, $p < 0.001$. These results indicated the dots number moderated the relationship between numerosity perception and mathematics ability.

## The correlations among other variables

The correlation between visual-spatial memory and mathematics ability reached marginal significance ($r = 0.26$, $p = 0.054$). Inhibition ability significantly correlated with $w_{small}$ ($r = -0.27$, $p = 0.044$) but not with $w_{large}$ ($r = -0.15$, $p = 0.273$). The correlations between all other variables also did not reach significance ($|rs| < 0.19$, $ps > 0.149$).

## Hierarchical regression analysis

To measure the extent to which mathematics ability could be predicted by numerosity perception of small numbers, we ran a hierarchical regression analysis with $w_{small}$ as a

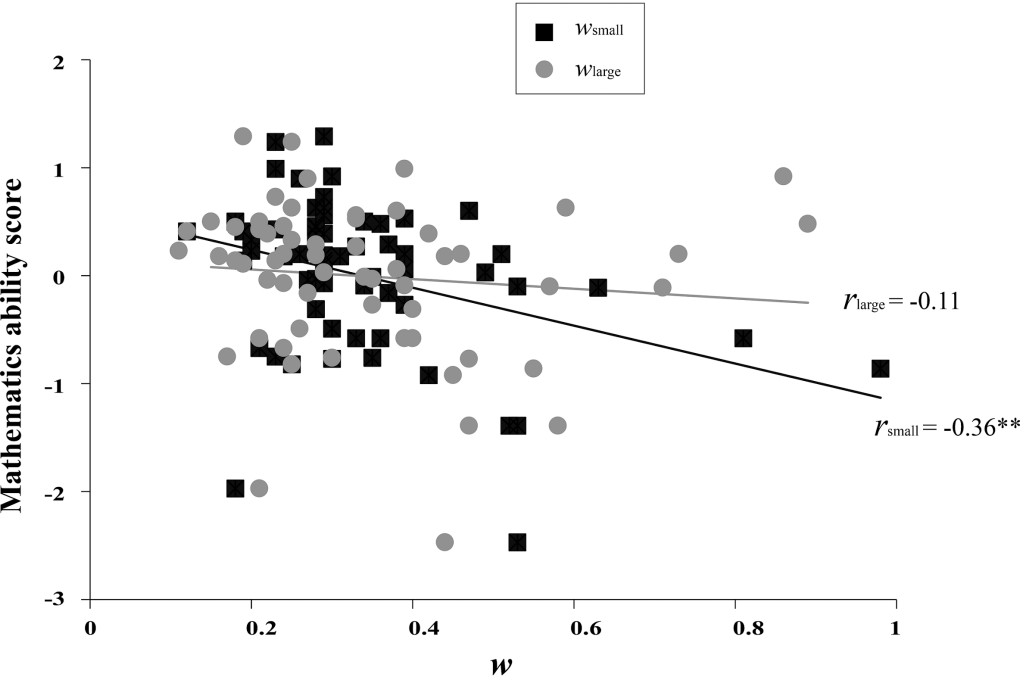

**Figure 1   The Pearson correlations between mathematics ability score and $ws$ ($w_{small}$ and $w_{large}$).** A filled symbol represents a participant's original data. See Table 1 for zero-order correlations. ** $p < 0.01$.

**Table 2   Hierarchical regression analysis.**

| Model | Predictor | $R^2$ | $R^2_{chang}$ | $F_{change}$ | df | P |
|---|---|---|---|---|---|---|
| First step | Visual-spatial memory, Inhibition ability, Set-switching ability, $w_{large}$ | 0.08 | —— | 1.24 | 56 | 0.304 |
| Second step | $w_{small}$ | 0.25 | 0.17 | 12.06 | 55 | 0.001 |

predictor and mathematics ability as the dependent variable. The $w_{large}$, visual-spatial memory, inhibition ability, and set-switching ability were entered as control variables. As shown in Table 2, all controlling variables only explained 8% of mathematics ability variance, $F_{change(56)} = 1.24$, $p = 0.304$. The numerosity perception of small numbers ($w_{small}$) still explained a significant portion of mathematics ability variance (17%), $F_{change(55)} = 12.06$, $p = 0.001$.

## DISCUSSION

The present study examined the relationship between numerosity perception and mathematics ability in adults and found the moderating role of dots number in this relationship. We showed that a significant association existed only between numerosity perception of small numbers and mathematics ability in adults. Even after controlling other cognitive variables, the association remained stable and robust.

Prior studies have not found a significant correlation between numerosity perception and mathematics ability in adults (*Gilmore et al., 2013*; *Inglis et al., 2011*). However, the inconsistency might be attributed to that many of these studies did not take into account the specific effect of the methodological differences in numerosity perception tasks and other cognitive abilities (*Dietrich et al., 2019*; *Fuhs & McNeil, 2013*; *Norris & Castronovo, 2016*). The present study found that the relationship between numerosity perception and mathematics ability might depend on the dots number. In addition, even after controlling other cognitive variables (*i.e.,* inhibition ability, visual-spatial memory, and set-switching ability), the numerosity perception of small numbers could still explain 17% of mathematics ability variance. These results were consistent with a similar study in children (*Anobile et al., 2016*), in which only small numerosity perception thresholds were significantly correlated with mathematics ability. Indeed, it is suggested that small numerosity perception was based on numerosity mechanism, and large numerosity perception was based on density mechanism (*Anobile, Cicchini & Burr, 2014*; *Zimmermann, 2018*). Thus, our findings supported the notion that dots number could influence the association between numerosity perception and mathematics ability in adults. Future studies with more levels of number of dots are required in order to further explore how dots number influences the relationship between numerosity perception and mathematics ability. In addition, our current findings may have particular practical implications for math learning. They can guide psychological interventions to help children and adults with math learning difficulties improve mathematics ability.

In this study, the role of the cognitive abilities in the relationship between numerosity perception and mathematics ability, particularly the inhibition ability, has not been validated. *Fuhs & McNeil (2013)* found correlations between numerosity perception, inhibition ability, and mathematics ability. They argued that the numerosity perception explained the mathematical achievement only when considering the contribution of inhibition ability. Our study only found correlations between numerosity perception of small numbers and mathematics ability but not between inhibition ability and mathematics skills. One possibility was that the participants' age influenced the relationship between the two constructs. Several studies supported the relationship between inhibition ability and mathematics ability in preschool and school-age children (*Bull & Scerif, 2001*). Yet, the nature of inhibition ability and mathematics ability profoundly changed with age (*Espy et al., 2004*). These variations inherently affected how inhibition ability related to mathematics ability (*Ahmed et al., 2019*). As discussed above, we speculated that the association of numerosity perception and mathematics ability might be stable and robust and exist in both children and adults; whereas the association of inhibition ability and mathematics ability might be diverse and labile and change with age. Future research could examine this issue.

Another critical finding pertained the selective relationship between inhibition ability and numerosity perception of small numbers. The results implied that adults with high levels of inhibition ability could better cope with small numerosity perception tasks. Adults needed to exact abstract numerical properties and inhibit extraneous continuous properties in small number discrimination tasks. But they only needed to exact continuous properties

to complete large number discrimination tasks (*Fuhs & McNeil, 2013*). It was not surprising that inhibition ability played a key role in numerosity perception of small numbers. In addition, although the correlation between visual-spatial memory and mathematics ability was significant in children (*Caviola, Mammarella & Dénes Szcs, 2020*), the present study found a marginal association in adults. We speculated that the relationship between visual-spatial memory and mathematics ability would be much less pronounced in adults since they had many years of schooling.

It was also important to note that the numerosity perception performances across two conditions were similar in the present study. These findings contradicted with the results of *Anobile, Cicchini & Burr (2014)*, who found that the numerosity perception threshold of small numbers was lower compared to large numbers. These inconsistencies could be partially attributed to individual differences in numerosity perception performance (*Halberda, Mazzocco & Feigenson, 2008*; *Ross, 2003*). For example, *Ross (2003)* used numbers from 8 to 64 as base numbers and asked participants to complete a numerosity discrimination task. Four of five observers' thresholds initially increased almost linearly with base numbers and decreased with base numbers of 30 or above. However, the thresholds of one observer were consistently higher than any of the others and increased with base numbers. The author suggested that there might not be a universally available numerosity discrimination pattern. Given the large individual differences in patterns of numerosity perception, the differences in numerosity perception performances between the small and large number conditions might be cancelled out in the present study. Despite the lack of significant differences between small and large numerosity perception, only small numerosity perception correlated with mathematics ability and inhibition ability. These results implied that there might be separate mechanisms for small and large numbers, but more sensitive indexes other than the numerosity perception threshold were needed to detect the difference between small and large number perception. In addition, it appeared likely that the difference between psychophysical methods used by the previous study and the current study to measure $w$ could explain the inconsistent findings. Specifically, *Anobile, Cicchini & Burr (2014)* used the adaptive Quest method, and we used the Method of Constant Stimuli. Future studies should compare the two different methods.

## CONCLUSIONS

This article demonstrated the relationship between numerosity perception and mathematics ability in adults and the influence of dots number on this relationship. The results suggested that mathematics ability was significantly associated with the numerosity perception of small numbers rather than large numbers. It seemed likely that the numerosity perception of small and large numbers were encoded differently and had a distinct effect on mathematics ability. Future studies should consider other potential domain-specific factors, such as SNS, to further explore the relationship between numerosity perception and mathematics ability. In addition, the neural mechanisms underlying small and large numerosity perception might also provide strong evidence for the effect of dots number and should be explored further.

## ACKNOWLEDGEMENTS

We would like to thank our participants who devoted their time to support this research and Feng Kong, Annebella Tsz Ho Choi, Zumei Wu, Yuezhen Li, Yingchun Wu, Qinglin Zhao for their assistance in data collection and manuscript preparation.

### Funding

The project was supported by the National Natural Science Foundation of China (NO.81671065) and the Department of Education of Guizhou Province (NO.2018520103). The funders had no role in study design, data collection and analysis, decision to publish, or preparation of the manuscript.

### Grant Disclosures

The following grant information was disclosed by the authors:
The National Natural Science Foundation of China: NO.81671065.
The Department of Education of Guizhou Province: NO.2018520103.

### Competing Interests

The authors declare there are no competing interests.

### Author Contributions

- Ji Sun conceived and designed the experiments, performed the experiments, analyzed the data, prepared figures and/or tables, authored or reviewed drafts of the paper, and approved the final draft.
- Pei Sun conceived and designed the experiments, analyzed the data, prepared figures and/or tables, authored or reviewed drafts of the paper, and approved the final draft.

### Human Ethics

The following information was supplied relating to ethical approvals (*i.e.,* approving body and any reference numbers):

This study was approved by the ethics committee of the School of Education Science, Anshun university (ID number: ASU-JYXY-201903).

### Data Availability

The raw measurements are available in the Supplementary File.

### Supplemental Information

Supplemental information for this article can be found online at http://dx.doi.org/10.7717/peerj.12660#supplemental-information.

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
