# Peer review of "The relationship between numerosity perception and mathematics ability in adults: the moderating role of dots number"

_PeerJ, doi:10.7717/peerj.12660_

## Round 0.1 · original submission · Major Revisions

As you can see, the Reviewers appreciate your paper but also asked for some major revision. Please address carefully their recommendation and I would be happy to reconsider your paper.

·

Basic reporting

Dear authors,
I am pleased to review a manuscript that I consider very informative in the field of numerical cognition. I appreciate the motivation of filling a very specific gap with important theoretical and educational contribution. I recommend this article for publication but only after a major revision motivated by the comments reported below, in the target sections. In particular, I ask you to address points related to introduction, method and discussion. Moreover, I suggest you to improve formal aspects related to the article structure and language. Finally, I encourage you to add a figure as a graphic visualization of the main findings.

BACKGROUND: The introduction needs more detail. I suggest you to further elaborate the paragraph on predictors of math abilities (lines 36-45). An informative organization could consider domain-general predictors (memory, EC and language) and domain-specific predictors. This last domain should contemplate not only the Approximate Number System (ANS) but also the Symbolic Number Comparison (SNC): Indeed, it has been recently shown how during development, the SNC becomes the better domain-specific predictor of math abilities (Caviola et al., 2020, DOI: 10.1111/desc.12957) and of ANS itself (Lyons et al., 2018, doi.org/10.1037/dev0000445; Lau et al., 2021, doi.org/10.1037/dev0001158). Even if the inclusion of the SNC as task in your method would have allowed more comprehensive conclusions; at least in the introduction, its role should not be overlooked.

ENGLISH LANGUAGE: The English language should be improved. Some examples include lines 81-83, 102, 160 and the long period from line 198 to line 202. The current phrasing does not affect comprehension but gives the impression that the article was written under time pressure, without an accurate checking. In line 106, the subtitle “Measures” is not appropriate and should be replaced with “Apparatus and software”.

ARTICLE STRUCTURE: The paragraphs are correctly divided, but more subtitles would help the reader better organize the material. For instance, the four paragraphs belonging to the Result section could be labelled as follows: “Math scores”, “Math abilities and domain-specific skills”, “Math abilities and domain-general skills”, “Math abilities and small numerosity perception”.

FIGURE: I encourage you to add a figure as a graphic visualization of the main findings.

Experimental design

AIMS AND SCOPE: The relevance of the research questions in the field of education and rehabilitation should be highlighted to make the article more in line with the scope of the target Journal.

METHOD: The method should be described with more details to ensure replicability. I list below relevant information, currently missing in the manuscript.

Apparatus: Does the setting include the use of a chin rest to ensure standard and steady posture of all participants across the different trials?

Procedure: Which was the order in which participants performed the different tasks?

Numerosity perception:
- Were the small and large number conditions blocked (as stated in line 87) and also counterbalanced?
- Were the patches of equal size? What size were they?
- How do you motivate the administration of so few practice trials compared to Anobile et al., 2014?

Domain-general skills: Since the “Set-switching ability” paragraph starts with definition of the ability, object of interest, and proceeds with description of task used to assess it, I suggest you to keep the same structure with all the domain-general skills by adding a short definition.

Validity of the findings

CONCLUSIONS: The conclusion is concise but not comprehensive. I would appreciate if you could add interpretation of the marginal correlation between visual-spatial memory and math ability and, most importantly, of the selective correlation between inhibition ability and discrimination of small numerosity. As a speculation, inhibition could help extract abstract numerical properties from continuous properties.

IMPLICATION: Implications in the field of education and rehabilitation would make the article more interesting for the readers of the Journal.

·

Basic reporting

In the current study, the authors asked whether numerosity is a predictor of formal mathematical ability in adults. The results showed a significant correlation but only when the numerosity perception was tested with relatively low numerosities. The study replicates and extends what was previously shown in a study with children.
The study is interesting and well conducted. I have a few considerations (which I list below), related to minor methodological issues and more importantly on how the results were shown.

Experimental design

Methods:
Stimuli have been selected with the Method of Constant stimuli that varied the number of dots from 50% to 200%. Into which and how many steps has this interval been divided (would be probably good to list the numerosity used)? I believe it is important to know for two reasons: 1) for replicability 2) it differs from some studies (including Anobile et al 2014 psy science and Anobile et al 2016 Dev psych) where the QUEST adaptive method was used instead. I think it is also worth discussing the possibility that this methodological difference may have led to the difference between the studies in the value of the w found between high and low numerosities.

Validity of the findings

Results:

1) Given that mathematics was measured with several subtests, and given that it has previously been shown that not all mathematical tasks equally correlate with numerosity estimation (at least in children , Anobile, Sievano, Burr 2013), it would be interesting to also show the correlations between w and the different math subtests (for both low and high numerosities separately).


2) While I have nothing against showing results by means of tables, I advise the authors to also show the data by means of correlation graphs, at least for zero-order correlations between w and maths (separately for low and high numerosities). Besides facilitating the understanding of the main results, it is also important because it allows to show individual subjects data, which otherwise would never be visible in the current form of the ms.

---

## Round 0.2 · accepted · Accept

It seems to me that you have addressed adequately all comments and I am happy to accept your paper.

·

Basic reporting

Dear Editor Giorgio Vallortigara and authors Sun Ji and Sun Pei,
even if in the reply to reviewers the reference to the modified or added lines did not match the ones in the revised article, it was possible to appreciate the revision process. The authors replied to each points raised by the reviewers in a comprehensive way. Therefore, I am glad to endorse this version of the manuscript for publication and I congratulate the authors for the important contribution to the research and the editor for his efficient job.

Yours faithfully,

Arianna Felisatti

Experimental design

no comment

Validity of the findings

no comment

·

Basic reporting

The authors did a great job with the revision. I have no further comments.

Experimental design

The authors did a great job with the revision. I have no further comments.

Validity of the findings

The authors did a great job with the revision. I have no further comments.

Additional comments

The authors did a great job with the revision. I have no further comments.